# The Expression of TP53-Induced Glycolysis and Apoptosis Regulator (TIGAR) Can Be Controlled by the Antioxidant Orchestrator NRF2 in Human Carcinoma Cells

**DOI:** 10.3390/ijms23031905

**Published:** 2022-02-08

**Authors:** Helga Simon-Molas, Cristina Sánchez-de-Diego, Àurea Navarro-Sabaté, Esther Castaño, Francesc Ventura, Ramon Bartrons, Anna Manzano

**Affiliations:** 1Departament de Ciències Fisiològiques, Facultat de Medicina i Ciències de la Salut, Universitat de Barcelona, L’Hospitalet de Llobregat, 08907 Barcelona, Spain; hsimon@ub.edu (H.S.-M.); csanchezdg@gmail.com (C.S.-d.-D.); fventura@ub.edu (F.V.); rbartrons@ub.edu (R.B.); 2Department of Experimental Immunology, Amsterdam UMC, 1105 AZ Amsterdam, The Netherlands; 3Department of Hematology, Amsterdam UMC, 1105 AZ Amsterdam, The Netherlands; 4University of Wisconsin Carbone Cancer Center, Wisconsin Institutes for Medical Research, Madison, WI 53705, USA; 5Departament d’Infermeria Fonamental i Medicoquirúrgica, Facultat de Medicina i Ciències de la Salut, Universitat de Barcelona, L’Hospitalet de Llobregat, 08907 Barcelona, Spain; aureanavarro@ub.edu; 6Centres Científics i Tecnològics, Universitat de Barcelona, L’Hospitalet de Llobregat, 08907 Barcelona, Spain; mcastano@ub.edu

**Keywords:** *TIGAR*, NRF2, metabolism, oxidative stress, cancer

## Abstract

Hyperactivation of the KEAP1-NRF2 axis is a common molecular trait in carcinomas from different origin. The transcriptional program induced by NRF2 involves antioxidant and metabolic genes that render cancer cells more capable of dealing with oxidative stress. The TP53-Induced Glycolysis and Apoptosis Regulator (*TIGAR*) is an important regulator of glycolysis and the pentose phosphate pathway that was described as a p53 response gene, yet *TIGAR* expression is detected in p53-null tumors. In this study we investigated the role of NRF2 in the regulation of *TIGAR* in human carcinoma cell lines. Exposure of carcinoma cells to electrophilic molecules or overexpression of NRF2 significantly increased expression of *TIGAR*, in parallel to the known NRF2 target genes *NQO1* and *G6PD*. The same was observed in *TP53*KO cells, indicating that NRF2-mediated regulation of *TIGAR* is p53-independent. Accordingly, downregulation of NRF2 decreased the expression of *TIGAR* in carcinoma cell lines from different origin. As NRF2 is essential in the bone, we used mouse primary osteoblasts to corroborate our findings. The antioxidant response elements for NRF2 binding to the promoter of human and mouse *TIGAR* were described. This study provides the first evidence that NRF2 controls the expression of *TIGAR* at the transcriptional level.

## 1. Introduction

The maintenance of reactive oxygen species (ROS) homeostasis is a critical determinant of cell survival. In proliferating tissues, a decreased supply of oxygen and nutrients and deregulated metabolic pathways usually pose a threat to the redox balance [1]. The role played by ROS in tumor initiation and development is controversial and has been shown to depend on many factors, including tumor origin, developmental stage, and tissue microenvironment. Moreover, an increasing body of evidence indicates that cancer cells adapt to an imbalanced redox status by developing alternative metabolic reactions or by strengthening existing ones. This allows them to balance ROS according to their needs and renders them insensitive to further stress inducers such as chemotherapy and radiation [1].

The TP53-induced glycolysis and apoptosis regulator (*TIGAR*) is an important gene coding for a regulatory enzyme of glycolysis and ROS detoxification [2]. The activity of this enzyme that has attracted most research interest is the dephosphorylation of fructose-2,6-bisphosphate (Fru-2,6-P_2_), the most potent allosteric activator of 6-phosphofructo-1-kinase (PFK-1) and an inhibitor of fructose 1,6-bisphosphatase [3,4]. In so doing, TIGAR increases the flux from fructose-6-phosphate to glucose-6-phosphate, fueling the pentose phosphate pathway (PPP). Therefore, TIGAR contributes to the production of nicotinamide adenine dinucleotide phosphate (NADPH) for glutathione (GSH) regeneration. Importantly, the capacity of TIGAR to increase the NADPH/NADP+ ratio and confer protection from ROS-induced apoptosis was found to be dependent on the activity of glucose-6-phosphate dehydrogenase (G6PD) [2]. This indicates that TIGAR constitutes a link between glycolysis and the PPP. In physiological conditions, TIGAR expression confers cytoprotection and can prevent tumor development, as shown in mouse intestinal epithelia [5]. However, in a neoplastic context, TIGAR helps increase the antioxidant capacity of malignant cells and favors tumor growth. This phenomenon has been described in different cancer cell lines and tumor types [4,6,7]. Consequently, the downregulation of TIGAR in tumors has been proposed as an appropriate strategy to increase the effectiveness of antineoplastic therapies based on ROS-mediated cell death [8,9]. Recently, the cytotoxic effects of the DNA methyltransferase inhibitor decitabine and the inhibitor of heme-oxygenase tin mesoporphyrin have been linked to decreased expression of TIGAR and reduced PPP in myeloid leukemia and non-small cell lung carcinoma (NSCLC), respectively [10,11].

One of the most relevant players in the cellular antioxidant response is the transcription factor nuclear factor (erythroid-derived 2)-like 2 (NRF2), encoded by the gene *NFE2L2* [12]. Through the activation of a wide range of detoxifying enzymes and metabolic effectors, NRF2 maintains ROS homeostasis and tissue integrity [13]. Under normal conditions, NRF2 is subjected to rapid turnover in the cytoplasm due to its binding to the ubiquitin ligase Kelch-like ECH-associated protein 1 (KEAP1) and Cullin 3 (CUL3). CUL3 ubiquitinates NRF2 and promotes its proteasomal degradation in the cytoplasm [13]. Upon exposure to electrophilic molecules or ROS, cysteine residues in KEAP1 are oxidized resulting in a conformational change, which disrupts the KEAP1-NRF2 binding and renders NRF2 free to translocate to the nucleus [13]. NRF2 forms heterodimers with small musculoaponeurotic fibrosarcoma oncogene homologue (sMaf) proteins, which are primarily located in the nucleus. The NRF2-sMaf heterodimer binds to antioxidant response elements (AREs) in the promoter of different genes thereby enhancing transcription [14]. AREs are typically sequences of 5’-TGAC/GnnnGC-3’ that function as cis-regulatory elements or enhancers [15]. Constitutive activation of NRF2 is common in malignant cells regardless of ROS levels [16,17]. NRF2 target genes include enzymes with a wide range of functions: ROS detoxification such as NAD(P)H quinone dehydrogenase 1 (NQO1) and thioredoxin reductase 1 (TXNRD1), NADPH synthesis such as G6PD or malic enzyme 1 (ME-1), and glutathione metabolism such as the glutamate-cysteine ligase catalytic (GCLC) subunit, among others [18]. Far from having a restricted role as antioxidant, the pleiotropy of NRF2 makes it a key transcription factor in processes related to each of the hallmarks of cancer [19]. Importantly, analysis of genetic alterations from different cancer genome sequencing studies using cBioportal [20,21] revealed that *KEAP1*/*NFE2L2* mutations are highly frequent in NSCLC, endometrial cancer, ovarian cancer, nerve sheath tumor and various forms of squamous cell carcinoma including cervical cancer. Abnormalities of *KEAP1*/*NFE2L2* in NSCLC and are associated with poor clinical outcome [22]. In cervical carcinoma, *KEAP1* expression is significantly decreased in the cytoplasm of cancer cells compared with that of normal cervical epithelial cells, which increases NRF2 activity. Constitutive NRF2 activation is associated with advanced stages of this type of tumors [23].

*TIGAR* expression can be regulated by p53 through a binding site in the proximal region of the first intron of the gene [2]. However, TIGAR has been found overexpressed in cell lines and tumors in which *TP53* has been either lost or mutated [4,5]. In fact, the expression of mouse *Tigar* is independent of p53 and p73 [24]. Considering that *TP53* is mutated in more than 50% of tumors [25], there is a clear need to find other transcriptional mechanisms which can account for TIGAR modulation. To date, only the specificity protein 1 (SP1) and the cAMP response element-binding protein 1 (CREB1) have been described to bind to the promoter of *TIGAR* [26,27]. However, neither of them is intrinsically involved in the antioxidant program of cells. Recently, it has been shown that TIGAR expression is dynamically regulated during the development of pancreatic ductal adenocarcinoma, and that together with NRF2 it has a key role in balancing ROS levels during the carcinogenic process [28].

The main aim of the present study was to determine whether NRF2 can transcriptionally regulate the expression of *TIGAR* in human carcinoma cells.

## 2. Results

### 2.1. NRF2 Activators Increase the Transcription of TIGAR Independently of p53

Activation of the endogenous NRF2 pathway can be triggered by electrophilic molecules that modify cysteine residues in KEAP1 and allow NRF2 to translocate to the nucleus [29]. In this study, the NRF2 activators sulforaphane (SFN) and dimethyl fumarate (DMF) were used in vitro to assess the capacity of endogenous NRF2 to regulate TIGAR expression. The cervical cancer cell line HeLa was exposed to SFN or DMF for 24 h, and the activation of the KEAP1-NRF2 axis was evaluated at the transcriptional level. Both NRF2 inducers increased the expression of *NFE2L2*, *TIGAR* and the NRF2 target genes *G6PD* and *NQO1*, with SFN having a stronger effect (Figure 1A,B). To assess whether the observed upregulation of TIGAR expression by NRF2 was transcriptionally mediated, HeLa cells were pretreated with actinomycin-D (Act-D) for 1 h prior to addition of SFN or DMF. Cells were analyzed 24 h after treatment with the activators by RT-qPCR. Inhibition of transcription by Act-D significantly prevented the induction of *TIGAR*, *NQO1* and *NFE2L2* by SFN and DMF (Figure 1C,D). Overall, these results indicate that NRF2 controls the expression of TIGAR at the transcriptional level in a similar way as it controls the expression of the previously described NRF2 target genes *NQO1* and *G6PD*.

Given that p53 is a direct regulator of TIGAR transcription [2], we wanted to assess the involvement of p53 in the ability of NRF2 to control the expression of *TIGAR*. The colorectal carcinoma cell lines HCT116 40.16 and HCT116 379.2, with wild-type and KO *TP53*, respectively, were used for that purpose. The NRF2 inducer SFN significantly enhanced the expression of *TIGAR* in both cell lines, in parallel to increased expression of *G6PD* and *NQO1* (Figure 1E,F), indicating that p53 is not necessary for the NRF2-mediated antioxidant response that upregulates *TIGAR*.

### 2.2. Transient Overexpression and Downregulation of NRF2 Affects TIGAR Expression

To further investigate how NRF2 regulates TIGAR expression, HeLa cells were transfected with an *NFE2L2*-pcDNA3 overexpression plasmid to transiently increase NRF2 protein levels. The expression of *NFE2L2*, *TIGAR*, *G6PD* and *NQO1* were increased 24 h after NRF2 overexpression (Figure 2A). These results were also confirmed at the protein level (Figure 2B,C). As a complementary approach, siRNA was used to determine the contribution of NRF2 to the basal expression of its target genes. Disruption of NRF2 expression caused a significant decrease in the mRNA levels of *G6PD*, *NQO1* and *TIGAR* (Figure 2D). Importantly, while the expression of *G6PD* and *NQO1* was reduced by more than 70% by the NRF2-targeting siRNA, the expression of TIGAR was reduced around 40%, indicating that other regulatory mechanisms beyond NRF2 are involved in the maintenance of basal *TIGAR* expression. We observed similar results at the protein level, showing again that although NRF2 depletion significantly reduces TIGAR protein levels, around 50% of the amount of TIGAR protein remains stable in NRF2-downregulated cells (Figure 2E,F). These observations indicate that the transcriptional regulation of NRF2 target genes is different; for some of them such as *G6PD* and *NQO1*, NRF2 is the main transcription factor, whereas for *TIGAR* a more complex regulatory machinery keeps both mRNA and protein levels more stable. 

Given the clinical relevance of constitutive activation of the KEAP1-NRF2 axis in NSCLC [22], we also performed NRF2 downregulation experiments in two lung adenocarcinoma cell lines, A549 and H460, which have high NRF2 basal levels due to missense and truncating mutations in *KEAP1*. In both cell lines, significantly reduced levels of TIGAR were detected upon NRF2 silencing (Figure 2G,H), confirming the results obtained in HeLa cells. 

### 2.3. Antioxidant Response Elements Are Present in the Promoter of TIGAR 

An in-silico analysis was performed to analyze the presence of putative NRF2 binding sites in the promoter of the human *TIGAR* gene. The ARE consensus sequence TGAC/GnnAGC, which has been reported to be the most conserved motif in the promoter of NRF2 target genes [30], was tracked along *TIGAR* promoter. Two putative NRF2 binding motifs were identified at −1048, −1040 and −1441, −1433 bp from the transcription start site (TSS) and were named ARE1 and ARE2, respectively, with ARE1 being the closest to the TSS. The Evolutionary Conserved Regions (ECR) Browser (available at http://ecrbrowser.dcode.org, accessed on 10 January 2022) [31] was used to determine the degree of conservation of the AREs identified across species. Both AREs were found to be conserved between human and Pan troglodytes while ARE2 was also found to be conserved in rhesus macaque (Figure 3). The sequence studied also comprised important elements for *TIGAR* expression, including the SP1 [26] and CREB1 [27] binding sites in the promoter, and the p53 binding site [2], in the first intron (Figure 3).

### 2.4. NRF2 Binds to the Promoter of Human TIGAR through an Antioxidant Response Element

The functionality of the AREs in the control of *TIGAR* expression by NRF2 was evaluated. Three different luciferase reporter constructs were generated containing both ARE1 and ARE2, or either ARE1 or ARE2 alone. A schematic representation of the constructs is provided in Figure 4A. Each of the three constructs (ARE1, ARE2, ARE1 + ARE2) was transfected together with a β-galactosidase reporter plasmid into HeLa cells, together with an *NFE2L2* overexpressing plasmid or with 20 µM SFN treatment. The transcriptional activity of the constructs containing ARE1 alone or both AREs was significantly higher than that of the empty vector both in cells overexpressing NRF2 (Figure 4B) and cells treated with SFN (Figure 4C). The luciferase activity of the construct containing ARE2 alone was identical to that of the empty vector (Figure 4B,C), indicating that NRF2 regulates *TIGAR* expression through ARE1.

The specificity of binding of NRF2 to ARE1 of *TIGAR* promoter was assessed by chromatin immunoprecipitation (ChIP) assays. Chromatin fractions from HeLa cells expressing basal NRF2 levels and NRF2-depleted cells were obtained and immunoprecipitated using either an anti-NRF2 antibody or a nonspecific IgG as negative control. ARE1 in the promoter of *TIGAR* was interrogated with specific primers and the ARE responsible for NRF2-mediated *NQO1* regulation was used as a positive control [32]. The region flanking ARE1 in the promoter of *TIGAR* was enriched in anti-NRF2 immunoprecipitated fractions from cells expressing NRF2, compared to NRF2-depleted cells. In chromatin fractions obtained with nonspecific IgG, no differences were observed (Figure 4D). Binding of NRF2 to the ARE in the promoter of *NQO1* was confirmed as a positive control (Figure 4E). Taken together, luciferase and chromatin assays indicate that NRF2 transcriptionally controls *TIGAR* through ARE1 and possibly other motifs.

### 2.5. NRF2 Enhances the Expression of Tigar in Mouse Primary Cell Cultures

To corroborate our findings, we were interested in determining whether NRF2 could also control the expression of *TIGAR* in nontransformed cells. For that, we used mouse primary osteoblasts given that NRF2 has been described to be central in the antioxidant response in the bone. In this tissue, ROS homeostasis is required for maintaining a correct equilibrium between osteoblast and osteoclast activities [33,34]. Primary mouse osteoblasts were exposed to DMF and SFN for 48 h and the expression of NRF2 target genes was subsequently analyzed by RT-qPCR. Mouse *Tigar* mRNA expression was increased by both NRF2 inducers in parallel to several other NRF2 target genes, *Txnrd1*, *Nqo1* and *Gclc*, in mouse primary cells (Figure 5A,B). Given that ARE1 and ARE2 found in human *TIGAR* were not conserved in mice, we were interested in determining whether NRF2 could regulate *Tigar* expression through more distant enhancer elements in mouse cells. After analyzing 8000 bp before the TSS of mouse *Tigar*, a putative ARE conserved between mouse and human genomes was identified at −5677, −5669 bp from the TSS (Figure 5C). ChIP assays confirmed the binding of NRF2 to this ARE in osteoblasts treated with 5 µM DMF for 48 h (Figure 5D).

## 3. Discussion

Several studies have shown that expression levels of *TIGAR* are high in tumor samples from different origin [4]. The Gene Expression Profiling Interactive Analysis (GEPIA) [35], which collects data from The Cancer Genome Atlas (TCGA) and the Genotype-Tissue Expression (GTEx), shows that higher expression of *TIGAR* is related to lower overall survival in different cancer types, including cervical squamous cell carcinoma. According to cBioportal [20,21], the most common genetic alteration in *TIGAR* is amplification, but even in the tumor types in which the *TIGAR* DNA sequence is most altered, that is, uterine carcinosarcoma, genomic alterations are found in only 7% of the patients. Therefore, understanding the mechanisms that drive *TIGAR* overexpression in cancer cells is crucial to unveil how the antioxidant potential of this enzyme can be blocked. In this study, we have characterized the transcriptional regulation of *TIGAR* by the pleiotropic antioxidant factor NRF2 in cell lines and primary cells. *TIGAR* was initially identified as a p53-induced gene coding for a bisphosphatase enzyme that has activity on Fru-2,6-P_2_ [2]. Thereafter, most studies involving TIGAR have characterized the capacity of this enzyme to redirect the glycolytic flux to the PPP and increase the NADPH/NADP+ ratio [4]. NRF2 orchestrates an antioxidant and metabolic program within the cell that has been related to the resistance of ROS-induced cell death and adaptation to nutritional stress, and which has an impact on all the defined hallmarks of cancer [13,19]. We show here that NRF2 activation increases the expression of *TIGAR* in the cervical cancer HeLa cell line both in response to NRF2 overexpression and following exposure to the electrophilic molecules SFN and DMF, which induce the dissociation of NRF2 from KEAP1. Enhancement of TIGAR expression by SFN was also observed in the human colorectal carcinoma cell lines HCT116 40.16 and HCT116 379.2, which bear wild-type and null p53, respectively. These results demonstrate that *TIGAR* can be regulated by NRF2 in different cellular contexts regardless of p53 expression, despite being initially described as a p53 response gene [2]. This finding is in line with previous studies that have reported p53-independent expression of *TIGAR* [5]. NRF2 can be an important regulator of *TIGAR* in these models.

NRF2 downregulation by siRNA resulted in decreased TIGAR levels in HeLa, H460 and A549 cells. Accordingly, we have shown that NRF2 enhances the transcription of human *TIGAR* through an ARE located on its promoter at −1048 from the TSS, and that the amount of chromatin immunoprecipitated from this specific region is decreased in NRF2-depleted cells. However, the effect of NRF2 depletion on ChIP assays is more striking in the *NQO1* ARE than in *TIGAR* ARE1, suggesting that other motifs might also be involved in the regulation of *TIGAR* by NRF2. These results are consistent with the data obtained by RT-qPCR, which showed that the decrease in *NQO1* expression upon NRF2 depletion is more pronounced than that of *TIGAR*. Plausible explanations for that could be differential recruitment of sMaf proteins to the promoter of NRF2 target genes, or different chromatin availability throughout the genomic regions where these target genes are located due to epigenetic marks. In a previous study, these features were reported to compromise NRF2 activity on its target genes [14]. The participation of other antioxidant response elements, such as the distal motif involved in the control of mouse *Tigar*, or other transcription factors in the modulation of human *TIGAR* in response to NRF2 activation cannot be ruled out. Our results indicate that NRF2 can function as an enhancer of *TIGAR* expression under certain cellular conditions, but it is not responsible for the maintenance of the basal expression of this gene. The transcription factor SP1 was described as binding to the proximal promoter of *TIGAR* [26], in a region that is highly conserved between species, as evidenced in the in-silico analysis with the ECR Browser. Therefore, SP1 might account for the maintenance of *TIGAR* basal expression levels. 

In mouse primary cells cultured in vitro, the transcriptional regulation of *Tigar* by NRF2 involves an ARE located at −5.677 bp from the TSS. Increased binding of NRF2 to this ARE was observed in parallel to enhanced *Tigar* expression in response to the NRF2 inducers DMF and SFN in mouse osteoblasts. The process of differentiation of osteocytes in the bone is linked to a shift of the metabolic machinery from glycolysis towards oxidative phosphorylation [36]. Recently, increased mitochondrial biogenesis has been described as increasing ROS levels and NRF2 activity during osteocytogenesis [34]. The induction of Tigar by NRF2 in mouse osteoblasts might also play a role in the promotion of mitochondrial metabolism in this cell type. In breast carcinoma cells, *TIGAR* overexpression leads to increased oxygen consumption and ATP production in parallel to enhanced levels of the translocase of the outer mitochondrial membrane 20 (TOM20) [6]. 

Overall, we describe here the mechanism of control of *TIGAR* expression by NRF2, which can explain how *TIGAR* is upregulated in response to oxidative stress and electrophilic molecules in cancer cell lines and healthy primary cells regardless of the expression of p53 (Figure 6). This newly described mechanism of *TIGAR* regulation by NRF2 can be of special interest in human cell lines lacking p53, or in other species where *TIGAR* expression is independent of p53, such as mice [24].

The *KEAP1*/*NFE2L2* mutational status has been directly linked to the responsiveness of certain cancer types to radio- and chemotherapy [22]. Besides, NRF2 addiction has been shown to define the metabolic signature of specific types of tumors. For example, glutamine addiction and induction of the PPP in NSCLS [17,18,37]. Recently, it has been observed that TIGAR and NRF2 can fulfil similar functions in the initiation and dissemination of mouse pancreatic ductal adenocarcinoma. Expression of NRF2 and TIGAR is required for tumor initiation and for the development of established metastases, however loss of either one of these genes leads to increased ROS and metastatic potential [28]. Targeting the NRF2 axis has shown promising preclinical results in terms of sensitivity to therapy [18]. It would be interesting to analyze whether NRF2 inhibition is also beneficial for those cancer types in which TIGAR abrogation has been shown to increase sensibility to radio- and chemotherapy, the case, for example, of glioblastoma [8,38]. Whereas pharmacological inhibitors of the KEAP1-NRF2-sMAF pathway are currently being developed [18,39] and constitute potential antineoplastic drugs [40], no molecules have been proposed as potential inhibitors of TIGAR to date. The specific effects of NRF2 inhibitors that might be attributed to the downregulation of TIGAR have yet to be deciphered. Heterogenous modulation of NRF2 target genes should also be considered, as experiments performed by our group indicate that downregulation of NRF2 in cell lines of different origin does not always have the same effects at the transcriptional level. Thus, strategies based on the disruption of the genetic expression or the pharmacological inhibition of the KEAP1-NRF2 axis should first evaluate the consequences of this inhibition on NRF2 targets. The expression of certain NRF2 downstream effectors might be compensated by alternative mechanisms that, ultimately, could render cancer cells even more aggressive or resistant to therapies.

The establishment of a direct link between NRF2 and TIGAR is of relevance as it should help in anticipating the coordinated functions of these two proteins while, at the same time, provide additional information as to how metabolic and redox processes are interconnected. As it has become evident, metabolism is no longer seen as the group of reactions that provide cellular energy; rather, metabolism is also at the heart of ROS homeostasis, anabolism, and cellular division. The most frequently studied role of TIGAR is its antioxidant capacity due to the bisphosphatase activity on Fru-2,6-P_2_; however, reported activity of this enzyme on other glycolytic intermediates [41] and in mitochondrial function [6] should not be overlooked. The branching of glycolytic metabolites to anabolic pathways such as amino acid or nucleotide synthesis and the regulation of mitochondrial metabolism are important processes in tumor development, and NRF2 has been shown to control key enzymes in these pathways. Our work situates TIGAR in the wide landscape of NRF2-regulated pathways by demonstrating that NRF2 functions as an enhancer in the transcriptional control of human and mouse *TIGAR* gene.

## 4. Materials and Methods

### 4.1. Human Cell Lines

HeLa (cervical carcinoma), HCT116 40.16 and HCT116 379.2 (colorectal carcinoma) cell lines were purchased from the American Type Culture Collection (ATCC; Manassas, VA, USA). Lung adenocarcinoma cell lines H460 and A549 were kindly provided by the laboratories of Dr. Ricardo Pérez-Tomás and Dr. Jose Luis Rosa, respectively. All cells were cultured in high-glucose Dulbecco’s modified Eagle’s medium (DMEM; Biological Industries, Kibbutz Beit-Haemek, Israel, 01-055-1A-24) supplemented with 10% foetal bovine serum (FBS; Biological Industries, 04-007-1A), 2 mM glutamine (Biological Industries, 03-020-1B) and penicillin/streptomycin (100 U/mL and 100 μg/mL, respectively; Biological Industries, 03-031-1B). Elsewhere, this medium composition is referred to as complete DMEM. Cells were maintained at 37 °C, in a 5% CO_2_ atmosphere and a relative humidity of 70–80%.

### 4.2. Primary Mouse Osteoblasts

Primary osteoblasts were isolated from mice calvariae as previously described [36]. All procedures were approved by the Ethics Committee for Animal Experimentation of the Generalitat de Catalunya. Briefly, the calvariae were dissected from P1–P4 pups, and soft tissue was discarded. A total of five to eight calvariae were pooled and digested in α-Minimum Essential Medium Eagle Alpha Modification (α-MEM; SIGMA, St. Louis, MO, USA, M8042) containing trypsin (0.025%)/collagenase II (1 mg/mL; Thermo Fisher Scientific, Waltham, MA, USA, 17101015). The product of the first 5 minutes of digestion was discarded, while the product of a double 20-minute digestion was centrifuged and seeded on 60 mm culture plates. Cells were used between passages 1 to 4. Osteoblasts were cultured at 37 ºC in osteogenic media ((α-MEM with 10% FBS (Biological Industries, 04-007-1A), 2 mM glutamine (Biological Industries, 03-020-1B), 1 mM pyruvate (Biological Industries, 03-042-1B), 100 U/mL penicillin and 0.1 mg/mL streptomycin (Biological Industries, 03-031-1B), 50 µg/mL ascorbic acid and 4 mM β-glycerophosphate).

### 4.3. Transient Overexpression of NRF2

HeLa cells were plated at a density of 30% in six-well plates and the transfection procedure was performed the following day using Lipofectamine LTX (Invitrogen, Carlsbad, CA, USA, 15338-100) according to the manufacturer’s indications in Opti-MEM (Thermo Fisher Scientific, 11058021). One microgram of NC16 pcDNA3.1 Flag NRF2 (Addgene, Watertown, MA, USA, 36971) or the corresponding pcDNA3.1 empty vector (Invitrogen) was transfected into each well. After 4 h of transfection, complete DMEM was added to each well. Cellular extracts were collected after 24 h of plasmid transfection.

### 4.4. Transient Downregulation of NRF2

HeLa, H460, A549, HCT116 40.16 or HCT116 379.2 cells were plated at a density of 15% in six-well plates and the transfection procedure was performed the following day using Oligofectamine (Invitrogen, 12252-011) according to the manufacturer’s indications in Opti-MEM (Thermo Fisher Scientific, 11058021). The small-interfering RNA (siRNA) HSS107130 (Thermo Fisher Scientific) targeting exon 5 of *NFE2L2* (NM_006164.5) was used at 100 nM: 5’-CCAACCAGUUGACAGUGAACUCAUU-3’. Negative Stealth RNAi control, medium GC was used as negative control (Invitrogen, 12935300) and referred to as scrambled siRNA (Scr.). After 4 h of transfection, complete DMEM was added to each well. Cellular extracts were collected after 72 h of siRNA transfection.

### 4.5. NRF2 Inducers

Dimethylfumarate (DMF; SIGMA, 242926) and sulforaphane (SFN; SIGMA, S4441) were prepared with DMSO and used at a final concentration of 5 or 20 µM. Treatments were performed in the corresponding culture medium for each cell type. DMSO was used as vehicle control. DMF was freshly prepared for each experiment. Cells were plated at a density of 30% the day before treatment. Cell lines were treated for 24 h whereas primary osteoblasts were treated for 48 h with the indicated compounds.

### 4.6. Protein Extraction and Western Blot

Cells were lysed with whole cell lysis buffer containing 50 mM Tris–Cl, 1% SDS and 10% glycerol and protein concentration was determined with the Pierce BCA protein assay (Thermo Fisher Scientific, 23228 and 23224). Equal amounts of total protein extracts were analyzed in 12.5% (*w*/*v*) SDS–PAGE. Western blot was performed using specific antibodies for NRF2 (Santa Cruz Biotechnologies, Dallas, TX, USA, sc-722, 1:1000), G6PD (Abcam, Cambridge, UK, ab993, 1:500), TIGAR (Santa Cruz Biotechnologies, sc-67273, 1:1000) and α-tubulin (SIGMA, T8203, 1:4000). Peroxidase-conjugated secondary antibodies goat anti-rabbit and goat anti-α-mouse (Advansta, San Jose, CA, USA, R-050072-500 and R-050071-500, respectively) were used. Immunostaining was carried out using the ECL technique (Biological Industries, 20-500-500). Densitometric analysis was performed using Multi-Gauge v3.0 (FujiFilm Corporation, Tokyo, Japan, v. 2007) software. Protein levels were normalized to α-tubulin in all experiments. 

### 4.7. RTqPCR Analysis

For RNA extraction, cells were maintained on ice and lysed with TRIsure (Bioline, London, UK, BIO-38033). Extraction was performed according to the manufacturer’s indications and RNA was quantified with NanoDrop One (Thermo Fischer Scientific) and reverse transcribed using the High Capacity cDNA Reverse Transcription Kit (Thermo Fischer Scientific, 4368813). mRNA expression analysis was performed through RT-qPCR using SensiFAST™ Probe Hi-ROX (Bioline, BIO-82005) and the following TaqMan Assays (Thermo Fisher Scientific): human *NFE2L2* (Hs00975961_g1), *G6PD* (Hs00166169_m1), *NQO1* (Hs01045993_g1), *TIGAR* (Hs00608644_m1) and Glyceraldehyde 3-phosphate dehydrogenase (*GAPDH*) (Hs99999905_m1); and mouse *Nfe2l2* (Mm00477784_m1), *Txnrd1* (Mm00443675_m1), *Nqo1* (Mm01253561_m1), *Gclc* (Mm00802655_m1), *Tigar* (Mm01183137_m1) and TATA-box binding protein (*Tbp*) (Mm01277042_m1). The expression of each gene of interest was normalized to that of *GAPDH* in human cells and *Tbp* in mouse cells. 

### 4.8. In silico Analysis of Human and Mouse TIGAR Promoter

An in silico analysis was performed to identify putative AREs corresponding to the sequence TGAC/GnnnGC in the promoter of human and mouse *TIGAR*. The sequences tracked covered −1630, +472 bp and −8000, +1 of the TSS of human and mouse TIGAR, respectively. The reference for the TSS was the NCBI (human TIGAR mRNA: NM_020375.3, mouse Tigar mRNA: NM_177003.5). The ECR Browser (available at http://ecrbrowser.dcode.org, accessed on 10 January 2022) [31] was used to analyse the degree of conservation of the AREs identified across species by comparing the genome sequences of Homo sapiens, Pan troglodytes (panTro3), Rhesus macaque (rheMac2) and Mus musculus (mm10).

### 4.9. Generation of TIGAR Promoter Constructs

Two bacterial artificial chromosomes (BACs) of human chromosome 12 (RP11-177D20 and RP11-74J21) were provided by the BACPAC Service of the Children’s Hospital Oakland Research Institute (ChORI, Oakland, CA, USA) as agar-stab culture. Specific information about the clones is available at bacpacresources.org. Bacteria were cultured and amplified in Luria Bertani (LB) medium and BACs were purified with a phenol/chloroform-based DNA extraction protocol. The region covering from −1623 to −635 from the TSS of TIGAR, containing the two AREs of interest, was amplified by PCR using MyFi™ DNA Polymerase (Bioline, BIO-21118) and the following primers FW: 5’-GCGTCCTTACAGATCTAGCATGG-3’ and RV: 5’-GCCCCTTGATAGCTAGCAAAGTTC-3’. The resulting 989 bp PCR product was cloned into the pCRR2.1-TOPO vector using the TOPO TA Cloning Kit (45-1641, Thermo Fisher Scientific) and then subcloned into the pGL3-Promoter Vector (Promega, Madison, WI, E1761), under the control of SV40 promoter, through double restriction enzyme digestion with SacI/XhoI (New England Biolabs, Ipswich, MA, USA, R0156S and R0146S, respectively). To generate the constructs containing each of the ARE, the pGL3-Promoter Vector containing the 989 bp construct was further digested with SmaI (New England Biolabs, R0141S)/SacI or SmaI/XhoI to generate a 467 bp construct containing only ARE1 (−1101, −635 bp) and a 522 bp construct containing only ARE2 (−1623, −1101 bp), respectively. The single ARE constructs were subcloned again in pGL3-Promoter Vectors. For all cloning procedures, JM109 competent cells (Promega, L2005) were used, cultured in LB agar plates and LB liquid culture containing 50 µg/mL ampicillin and purified with GeneElute Plasmid Miniprep or Maxiprep Kits (SIGMA, PLN350 and NA0410, respectively). Phosphatase FastAP (Thermo Fisher Scientific, EF0651) and T4 DNA Ligase (Thermo Fisher Scientific, EL0011) were used to dephosphorylate 5’-P termini of vectors and to perform ligations, respectively. Specific PCR, digestion and ligation conditions are available on request.

### 4.10. Luciferase Assays

HeLa cells were plated at a density of 30% in six-well plates and the following day 0.8 µg of the corresponding luciferase reporter construct was transfected to each well using Lipofectamine LTX (Invitrogen, 15338-100) according to the manufacturer’s indications in Opti-MEM (Thermo Fisher Scientific, 11058021). The constructs used were pGL3-Promoter Vector containing the 989 bp with both AREs of *TIGAR* promoter (ARE1 + ARE2), pGL3-Promoter Vector containing only ARE1 (ARE1), pGL3-Promoter Vector containing only ARE2 (ARE2) or the corresponding pGL3-Promoter empty vector (Promega, E1761). To assess the efficiency of the transfection and to normalize luciferase expression values, 0.5 µg of RSV-β-galactosidase plasmid was transfected into each well. After 4h of transfection, cells were trypsinized and split (each well of a six-well plate was split into three wells of a 12-well plate) and complete DMEM was added to each well. Luciferase experiments were performed in cells overexpressing NRF2 or in cells treated with SFN. For NRF2 overexpression, 1 µg of NC16 pcDNA3.1 Flag NRF2 (Addgene, 36971) was co-transfected with the luciferase and β-galactosidase plasmids. For SFN treatment, SFN was added to a final concentration of 20 µM when cells were split into 12-well plates. After 24 h of transfection, cells were washed twice in cold PBS and lysed in buffer of the Luciferase Assay System (Promega, E1500). Fifty microliters of cell lysate were used to measure luciferase activity. β-Galactosidase activity was determined in 30 μL of cell lysate using the luminescent β-galactosidase detection kit II (Takara Bio, Shiga, Japan, 631712). Both activities were measured in a TD 20/20 luminometer (Turner Designs, San Jose, CA, USA) in acquisition intervals of 10 and 5 s, respectively.

### 4.11. Chromatin Immunoprecipitation (ChIP) Assays

HeLa and osteoblasts cells were plated at a density of 30% in 10 cm plates and either transfected with 6 µg of NC16 pcDNA3.1 Flag NRF2 (Addgene, 36971) or treated with DMF the following day. After 24 h, cells were fixed with 1% formaldehyde for 10 min, and the reaction was stopped with 0.25 M glycine for 5 min. Cells were maintained on ice, lysed with a buffer containing 1% SDS, 10 mM EDTA and 50 mM Tris and sonicated to obtain chromatin fragments of 500–1000 bp in a Branson Sonifier 250 (Branson Ultrasonics, Brookfield, CT, USA) with six cycles of ten pulses at duty cycle 0.8 seconds and output 5. Two 10-cm plates were lysed for each experimental condition. ChIP was carried out using 7.6 µg of NRF2 antibody (Cell Signaling Technology, Danvers, MA, USA, 12721) or nonspecific rabbit IgG (Merck Millipore, Burlington, Massachusetts, 12-372) as a control and purified with 20 µL of Magna ChIP protein A+G magnetic beads (Merck Millipore, 16-663). The complexes were washed twice with four different buffers and eluted with a solution of 1% SDS and 0.1 M NaHCO3. Reversion of cross-linking was carried out by overnight incubation with 0.2 M NaCl at 65 °C, followed by treatment with proteinase K and RNase A. The DNA fragments were purified using the QIAquick gel extraction kit (Qiagen, Hilden, Germany, 28706) and analyzed by RT-qPCR. Input samples were prepared with 12.5% of the chromatin material used for an immunoprecipitation in each experimental condition. Inputs were analyzed in parallel and used to normalize chromatin enrichment. RTqPCR was performed with SYBR Select Master Mix (Thermo Fischer Scientific, 4472908) at 45 cycles with the following primers:

FW human TIGAR (5’-AATGGCGTGAACCCGGGAGGCG-3’)

RV human TIGAR (5’-GCCTTCAGAACGTTGAGGGAGTTGC-3’)

FW human NQO1 (5’- TGGCATGCACCCAGGGAAGTGTGT-3’)

RV human NQO1 (5’-AGCCGGATGCGGATTACTGTGGTGC-3’)

FW mouse Tigar (5’-AGTGACAGGCTAAACGGCCAGGCA-3’)

RV mouse Tigar (5’-AGCTGGGGGCGGAGGAAGATTGGTT-3’).

### 4.12. Statistical Analysis

Results are expressed as the mean ± SEM of independent experiments. Differences were analyzed with the corresponding statistical test, according to the nature of the data represented. The number of experiments as well as the statistical method applied are indicated in each figure. Significant differences at *p* < 0.05, 0.01, and 0.001 between conditions are indicated by ∗, ∗∗, and ∗∗∗, respectively. All calculations were performed using the GraphPad Prism version 4.00 for Windows (GraphPad Software, La Jolla, CA, USA).

## 5. Conclusions

In summary, the present study demonstrates for the first time that NRF2 activation, either by electrophilic molecules such as SFN or DMF, or by NRF2 overexpression, regulates the expression of the glycolytic and ROS regulator gene *TIGAR* through the AREs located in its promoter.

## Figures and Tables

**Figure 1 ijms-23-01905-f001:**
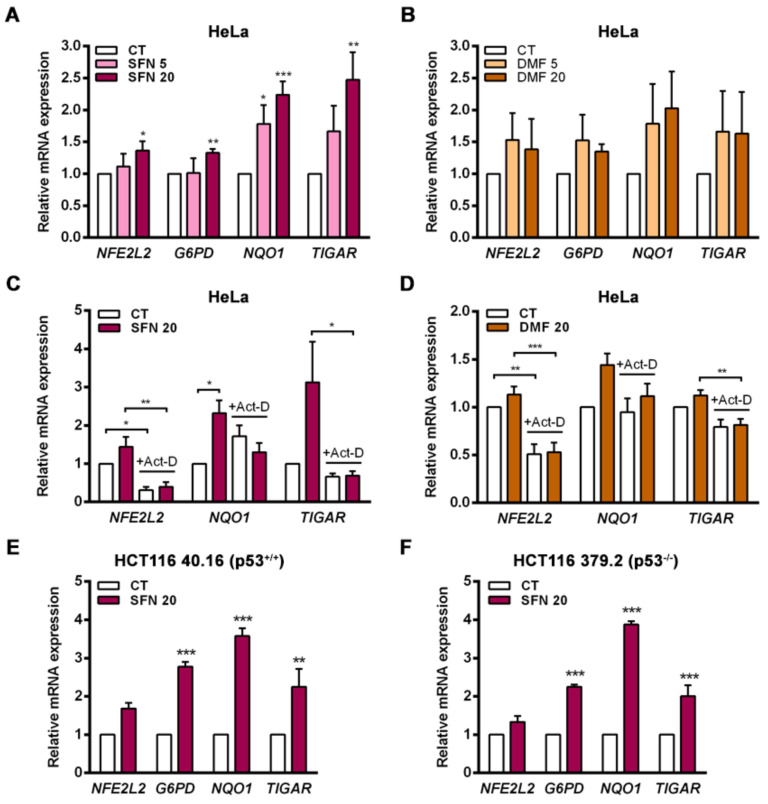
The expression of *TIGAR* is modulated by NRF2 activators in a transcription-dependent manner and independently of p53. (**A**) HeLa cells were treated with 5 or 20 µM (**A**) sulforaphane (SFN) (*n* = 4) or (**B**) dimethyl fumarate (DMF) (*n* = 5) for 24 h and analyzed by RT-qPCR. (**C**,**D**) HeLa cells were pretreated with 5 µg/mL actinomycin-D (Act-D) and then subsequently exposed to (**C**) 20 µM SFN (*n* = 4) or (**D**) 20 µM DMF (*n* = 5) for 24 h and analyzed by RT-qPCR. (**E**,**F**). (**E**) HCT116 40.16 (*n* = 3) and (**F**) HCT116 379.2 (*n* = 3) cell lines were treated with 20 µM sulforaphane (SFN) for 24 h and subsequently analyzed by RT-qPCR. Data are presented as the mean fold change relative to untreated cells (CT) ± SEM and differences were analyzed with (**A**,**B**) one-way ANOVA using the Holm–Sidak method for multiple comparisons, (**C**,**D**) two-way ANOVA using the Holm–Sidak method for multiple comparisons or (**E**,**F**) independent *t*-tests (* *p* < 0.05, ** *p* < 0.01, *** *p* < 0.001).

**Figure 2 ijms-23-01905-f002:**
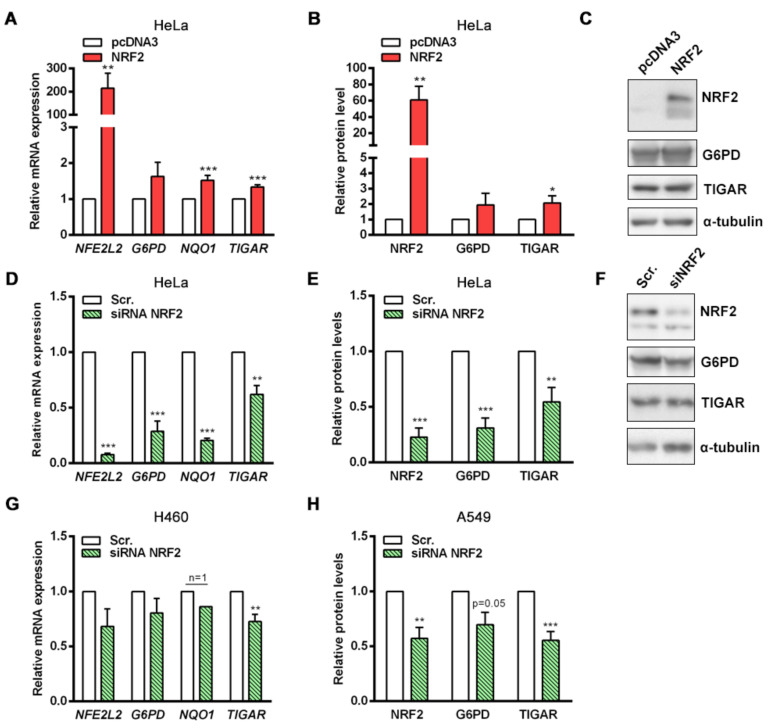
Specific modulation of NRF2 expression affects TIGAR levels. (**A**,**B**) HeLa cells were transfected with 1 µg of a pcDNA3 plasmid coding for human *NFE2L2* or the corresponding empty vector (pcDNA3) and subsequently analyzed after 24 h by (**A**) RT-qPCR (*n* = 14) or (**B**) Western blot (NRF2 and TIGAR, *n* = 6, G6PD *n* = 3). (**C**) Representative Western blot images of the conditions specified in B. (**D**,**E**) HeLa cells were transfected with 100 nM *NFE2L2*-targeting siRNA or the corresponding scrambled siRNA (Scr.) and analyzed after 72 h by (**D**) RT-qPCR (*n* = 5) or (**E**) Western blot (*n* = 5). (**F**) Representative Western blot images of the conditions specified in E. (**G**) H460 cells were transfected with 100 nM *NFE2L2*-targeting siRNA or the corresponding scrambled siRNA and analyzed after 72 h by RT-qPCR (*n* = 4). (**H**) A549 cells were transfected with 100 nM *NFE2L2*-targeting siRNA or the corresponding scrambled siRNA and analyzed after 72 h by Western blot (NRF2 *n* = 4, G6PD *n* = 3, TIGAR *n* = 8). Data are represented as the mean ± SEM relative to the corresponding control (pcDNA3 or Scr.) and differences were analyzed with independent *t*-tests (* *p* < 0.05, ** *p* < 0.01, *** *p* < 0.001).

**Figure 3 ijms-23-01905-f003:**
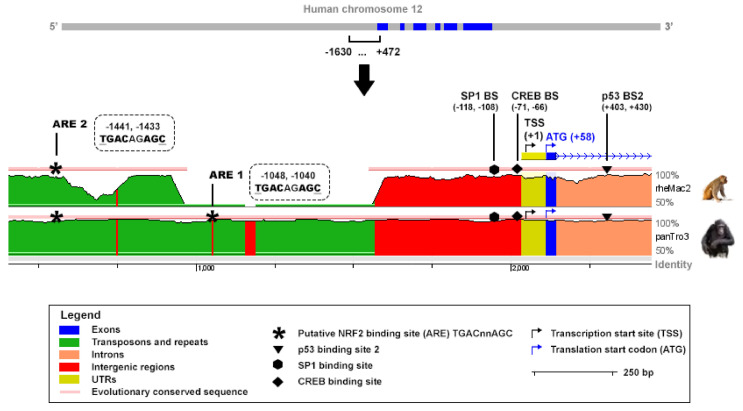
In-silico study of conserved AREs in the promoter of the human *TIGAR* gene. The ECR Browser (available at http://ecrbrowser.dcode.org, accessed on 10 January 2022) [31] was used to analyze the sequence from 4.428.749 to 4.430.851 bp of human chromosome 12 (−1630, +472 from the transcription start site (TSS)). The sequence was compared to the genome of Pan troglodytes (panTro3) and rhesus macaque (rheMac2). Horizontal red lines above the genomic sequences indicate evolutionary conserved regions. The ECR Browser image has been modified to highlight the most relevant elements in the present study. Antioxidant response elements (AREs) corresponding to TGACnnAGC motifs are represented by asterisks. The SP1, CREB1 and p53 binding sites are also indicated.

**Figure 4 ijms-23-01905-f004:**
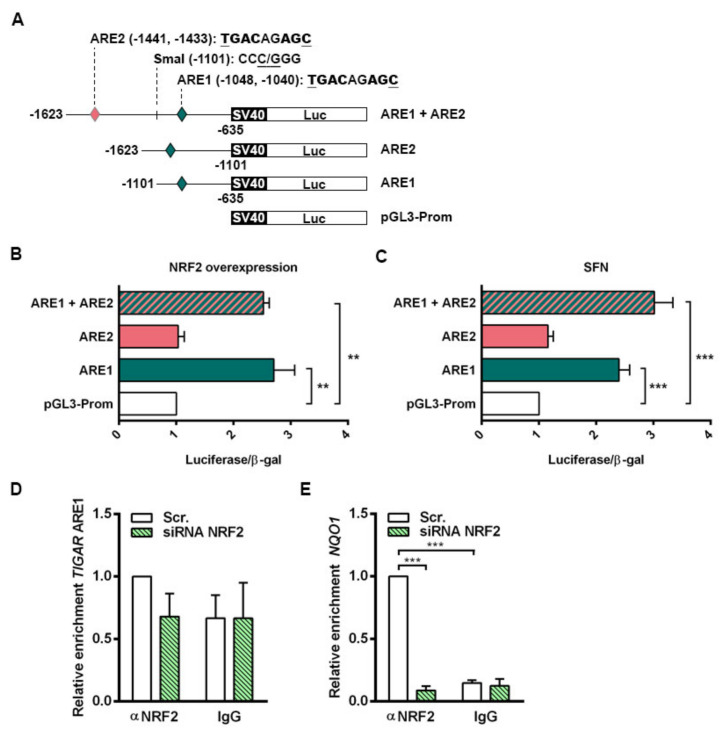
NRF2 enhances the transcription of human *TIGAR* through an ARE located on its promoter. (**A**) Schematic representation of the pGL3 promoter luciferase reporter constructs containing the antioxidant response elements (AREs) in human *TIGAR*: ARE1, ARE2 or both (ARE1 + ARE2). (**B**,**C**) Luciferase activity of the pGL3 luciferase reporter constructs normalized to β-galactosidase activity in (**B**) HeLa cells co-transfected with an *NFE2L2* overexpressing plasmid (*n* = 3) and (**C**) HeLa cells treated with 20 µM sulforaphane (SFN) (*n* = 6). Data are represented as the mean fold change relative to the corresponding control (empty pGL3 promoter vector (pGL3-Prom). (**D**,**E**) RT-qPCR analysis of *TIGAR* ARE1 (**D**) or *NQO1* ARE (**E**) in ChIP-enriched chromatin fractions from HeLa cells transfected with 100 nM *NFE2L2*-targeting siRNA or Scr. siRNA and analyzed after 72h. Specific antibody against NRF2 or nonspecific IgGs were used and chromatin enrichment was normalized to input fractions (*n* = 3). Data are represented as the mean fold change relative to the Scr. condition immunoprecipitated with anti-NRF2 ± SEM. Differences were analyzed with (**B**,**C**) one-way ANOVA or (**D**,**E**) two-way ANOVA using Tukey’s method for multiple comparisons (** *p* < 0.01, *** *p* < 0.001).

**Figure 5 ijms-23-01905-f005:**
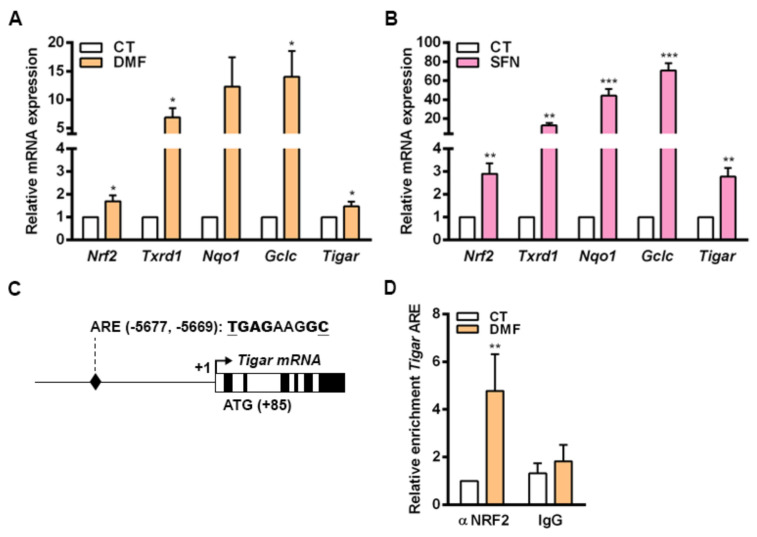
NRF2 regulates the expression of *Tigar* in mouse primary osteoblasts. (**A**,**B**) Primary mouse osteoblasts were treated with (**A**) 20 µM dimethylfumarate (DMF) or (**B**) 20 µM sulforaphane (SFN) for 48 h and the expression of *Nrf2*, *Txnrd1*, *Nqo1*, *Gclc* and *Tigar* was determined by RT-qPCR (*n* = 5). (**C**) Schematic representation of mouse *Tigar* promoter with the identified antioxidant response element (ARE). (**D**) Primary mouse osteoblasts were treated with 5 µM DMF for 48 h and ChIP was performed. ChIP-enriched chromatin of *Tigar* ARE from anti-NRF2 antibody or nonspecific IgGs fractions was analyzed by RT-qPCR and normalized to input fractions (*n* = 5). Data are represented as the mean ± SEM and differences were analyzed with (**A**,**B**) independent *t*-tests or (**D**) two-way ANOVA using Tukey’s method for multiple comparisons (* *p* < 0.05, ** *p* < 0.01, *** *p* < 0.001).

**Figure 6 ijms-23-01905-f006:**
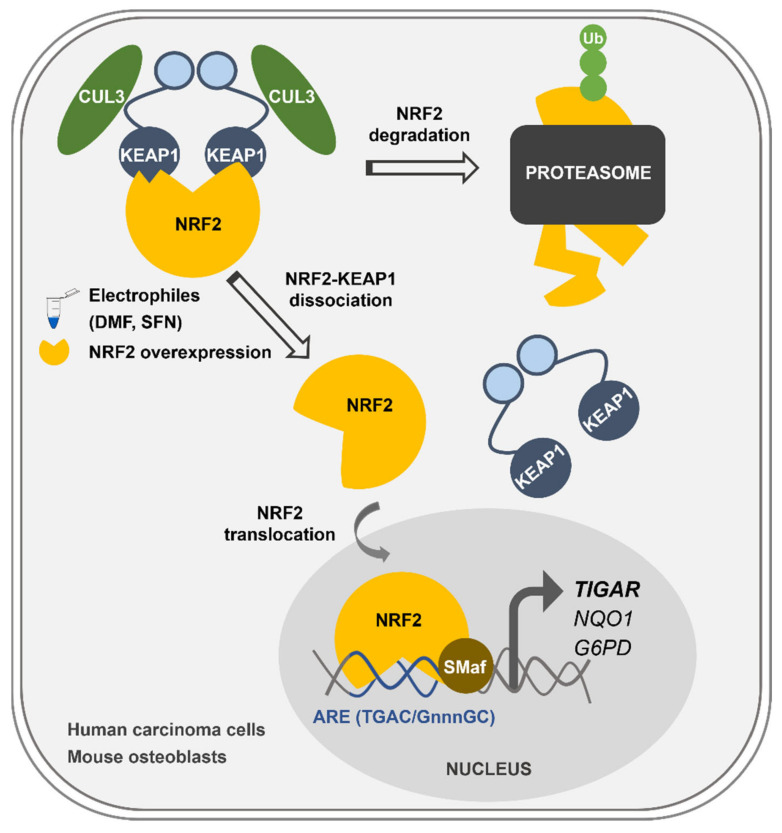
Graphical abstract of the main findings of this work. Transcriptional regulation of *TIGAR* by NRF2 in human and mouse cells. Upon exposure to the electrophilic molecules SFN and DMF, or after NRF2 overexpression, NRF2 is liberated from KEAP1 and translocates to the nucleus. NRF2 forms heterodimers with sMafs proteins in antioxidant response elements (AREs) located at the promoter of *TIGAR* and other antioxidant genes such as *NQO1* and *G6PD*, enhancing their transcription.

## Data Availability

The data presented in this study are available on request from the corresponding author.

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
