# Peer review of "The Expression of TP53-Induced Glycolysis and Apoptosis Regulator (TIGAR) Can Be Controlled by the Antioxidant Orchestrator NRF2 in Human Carcinoma Cells"

_ijms, 2022, doi:10.3390/ijms23031905_

Round 1

Reviewer 1 Report

Review of the manuscript entitled: “The expression of TP53-Induced Glycolysis and Apoptosis Regulator (TIGAR) can be controlled by the antioxidant orchestrator NRF2”. The manuscript is interesting. However, some corrections should be made.

Firstly, I think You could add a cell line name to the title.

The aim of the study should be clearly stated, for example “Therefore, the aim of present study was to…” or “The aim of present study was to…”.

References must be added after: “In proliferating tissues, however, a decreased supply of oxygen and nutrients, especially within the tumor mass, and deregulated metabolic pathways that support tumor growth usually pose a threat to the redox balance”; “One of the most relevant players in the cellular antioxidant response is the transcription factor nuclear factor (erythroid-derived 2)-like 2 (NRF2), encoded by the gene NFE2L2.”; “Under normal conditions, NRF2 is subjected to rapid turnover in the cytoplasm due to its binding to the ubiquitin ligase Kelch-like ECH-associated protein 1 (KEAP1) and Cullin 3 (CUL3), which ubiquitinates NRF2 and promotes its proteasomal degradation in the cytoplasm.”; “Thereafter, most studies involving TIGAR have characterised the capacity of this enzyme to redirect the glycolytic flux to the PPP and increase the NADPH/NADP+ ratio.”

Introduction contains many long sentences should be shortened (broken into smaller ones). If it is possible, please shorten the introduction because it is a bit too long.

All abbreviations when introduced should be explained first for example “NADPH”, “GSH” on page 2.

The unnecessary gap is left on page 5.

The GLB1 gene (β-galactosidase) is responsible for cellular aging and forms a receptor for extracellular matrix proteins – especially elastin derived peptides. Have the Authors considered this in the research?

As far as I am able to accept different human cell lines, housing the results with mice is perhaps an exaggeration?

A549 cells grow in F12K medium, why weren't they grown in such medium?

Author Response

Please see the attachment for our point-by-point answer to Reviewer 1.

Reviewer 2 Report

The manuscript entitled  The expression of TP53-Induced Glycolysis and Apoptosis Regulator (TIGAR) can be controlled by the antioxidant orchestrator NRF2 elucidated  the first evidence that NRF2 controls the expression of TIGAR at the transcriptional level . 

In previous study, the authors demonstrated that TIGAR is upregulated in stimulated human lymphocytes through the PI3K/AKT signaling pathway, which contributes to the redirection of the carbon flux to the PPP. In this study, the authors definitely showed the expression of TIGAR would be controlled by by the antioxidant orchestrator NRF2. 

I think it means an excellent study to demonstrates for the first time that NRF2 activation, either by electrophilic molecules such as SFN or DMF, or by NRF2 overexpression, regulates the expression of the glycolytic and ROS regulator gene TIGAR through AREs located in its promote. I think this is a useful article and I think it makes a potent scientific contribution in oxidative stress and metabolism. I would recommend the manuscript for minor revision.

Just a minor point,
Introduction must be improved by incorporating more recent references including cellular antioxidant response and TIGAR. 

Author Response

Please see the attachment for our point-by-point answer to Reviewer 2.

Round 2

Reviewer 1 Report

The Authors explained all doubts very well.